# Shaping Frontline Practices: A Scoping Review of Human Factors Implicated in Electrical Safety Incidents

**Tristan W. Casey [1,\*], Hannah M. Mason [2], Jasmine Huang [3] and Richard C. Franklin [2]**

[1] Safety Science Innovation Lab, Griffith University, Brisbane, QLD 4111, Australia

[2] College of Public Health, Medical and Veterinary Sciences, James Cook University, Townsville, QLD 4811, Australia; hannah.mason@jcu.edu.au (H.M.M.); richard.franklin@jcu.edu.au (R.C.F.)

[3] School of Applied Psychology, Griffith University, Brisbane, QLD 4111, Australia; jasmine.huang@griffithuni.edu.au

[\*] Correspondence: tristan.casey@griffith.edu.au

**Abstract:** Injuries sustained while performing electrical work are a significant threat to the health and safety of workers and occur frequently. In some jurisdictions, non-fatal serious incidents have increased in recent years. Although significant work has been carried out on electrical safety from a human factor perspective, reviews of this literature are sparse. Thus, the purpose of this review is to collate and summarize human factors implicated in electrical safety events. Articles were collected from three databases (Scopus, Web of Science, and Google Scholar), using the search terms: safety, electri\*, human factors, and arc flash. Titles and abstracts were screened, full-text reviews were conducted, and 18 articles were included in the final review. Quality checks were undertaken using the Mixed Methods Appraisal Tool and the Critical Appraisal Skills Program. Environmental, individual, team, organizational, and macro factors were identified in the literature as factors which shape frontline electrical worker behavior, highlighting the complexity of injury prevention. The key contributions of this paper include: (1) a holistic and integrated summary of human factors implicated in electrical safety events, (2) the application of an established theoretical model to explain dynamic forces implicated in electrical safety incidents, and (3) several practical implications and recommendations to improve electrical safety. It is recommended that this framework is used to develop and test future interventions at the individual, team, organizational, and regulator level to mitigate risk and create meaningful and sustainable change in the electrical safety space.

**Keywords:** arc flash; electrical safety; human factors; arc burn; electrical explosion

## 1. Introduction

Most electrical safety incidents are unforgiving in terms of their effects on human life. For instance, arc flash is an insidious hazard that involves current flowing through the air between phase conductors, or between phase conductors and the ground—essentially, an unexpected electrical short-circuit that produces an arc of electricity and can cause significant harm, including death [1]. Arc flash plasma temperatures can exceed 2800 °C to 19,000 °C, causing burns. Additionally, exploding materials can eject shrapnel and sound waves, resulting in additional severe injuries [2]. Other types of electrical safety events range from minor shocks through to electrocutions, often resulting in permanent nerve and tissue damage [3], and death.

Around the world, injuries in the electrical industry occur frequently. Global industry reports highlight the frequency and severity of electrical safety events, and in some countries, these incidents have increased. For instance, the US Bureau of Labor Statistics reported that in 2019 [4], 166 electrical industry fatalities occurred, which was the most since 2011 and a 3.75% increase from 2018. US non-fatal electrical injuries totaled 1900 in 2019 and the overall trend in injuries has been constant over the past decade [4]. In other

jurisdictions, such as Australia, electrical safety events such as arc flash seem to be increasing, with 32 Queensland-based arc flash incidents over a five-year period between 2013 and 2018 [5], and an estimated one person per month being admitted to New South Wales hospitals for electrical burn injuries [6].

Although the science of electrical safety from a technical engineering perspective continues to evolve [2], most safety incident investigations highlight the centrality of human factors. For instance, an incident report from the Queensland Mines Inspectorate [7] indicated that poor communication between electrical contractors (leading to misunderstandings of risk and work activities), low electrical safety knowledge, failure to detect environmental conditions, and several organizational factors such as the level of attention and investment in electrical safety were implicated in the incident. Recent international research has also identified the importance of human factors in underpinning electrical safety incidents, such as poor work planning, production pressure, poor risk awareness and complacency, and job competency [8]. Despite these findings, little work has been carried out to collate and synthesize the electrical safety human factors literature.

Accordingly, this scoping review aims to describe the most recent human factors electrical safety literature (i.e., from 2000 onwards). The first section briefly summarizes a dynamic safety management model of relevance to the electrical safety context. In the next section, we describe how the model can be used to interpret the results of our scoping review by relating different categories of human factors together in a dynamic and systems-oriented manner. In the final section, a discussion of electrical work injury prevention recommendations based upon insights from this model are described.

*The Role of Human Factors in Shaping Electrical Contractor Safety*

Human factors are recognized as core contributors to the risk of electrical safety incidents [9]. Broadly, human factors are the range of psychological, social, and work-related factors that interfere with the performance of people. Identified human factors that are specific to the electrical industry include: workplace culture (the unwritten rules for social conduct and sense-making), cognitive task demands such as information overload, and even psychological wellbeing and health issues [2].

Focusing on frontline worker practices is important to prevent unsafe electrical practices because ultimately, it is the operational safety decisions and (in)actions of workers that release hazards and cause injuries [10]. However, we must move beyond frontline human behavior and towards upstream factors if the deeper contributions are to be ameliorated [11]. The idea that broader team and organization factors shape frontline practices is not new and has progressed in maturity over the past 30–50 years. 'Sharp-end' behavior is not only the product of individual-level human factors such as personality, attitude, and emotional experiences, but also how these factors interact with phenomena such as the physical workspace and environment, social elements such as organizational culture and safety climate, and human capabilities, including knowledge, skills, and motivation [12]. Although most arc flash injuries are precipitated by unsafe acts such as failing to isolate energy sources or otherwise working live [13], not donning the required Personal Protective Equipment or PPE [14], and failing to recognize risk in some types of maintenance work [15], these acts are mostly unintentional or accepted work practices, and a symptom of deeper trouble within the work system, such as latent local or organizational factors. Thus, this review focuses on how broader systems factors, both internal and external to the organization, shape the self-protective and risk management behaviors of frontline electrical workers.

The performance environment in which electrical work takes place is appropriately represented by Rasmussen's [16] dynamic risk modeling framework. As shown by Figure 1, it summarizes the forces that act on work operations to drive it closer to non-safety, as well as the counter-pressure exerted by safety campaigns and initiatives (e.g., lock out tag out, awareness campaigns, supervision, and enforcement of standards). Rasmussen [16] argued that because of the efficiency and least-effort gradients, work nearly always ends

up migrating close to the boundary of acceptable performance, and in some cases, over the edge into the error margin and beyond (resulting in an incident).

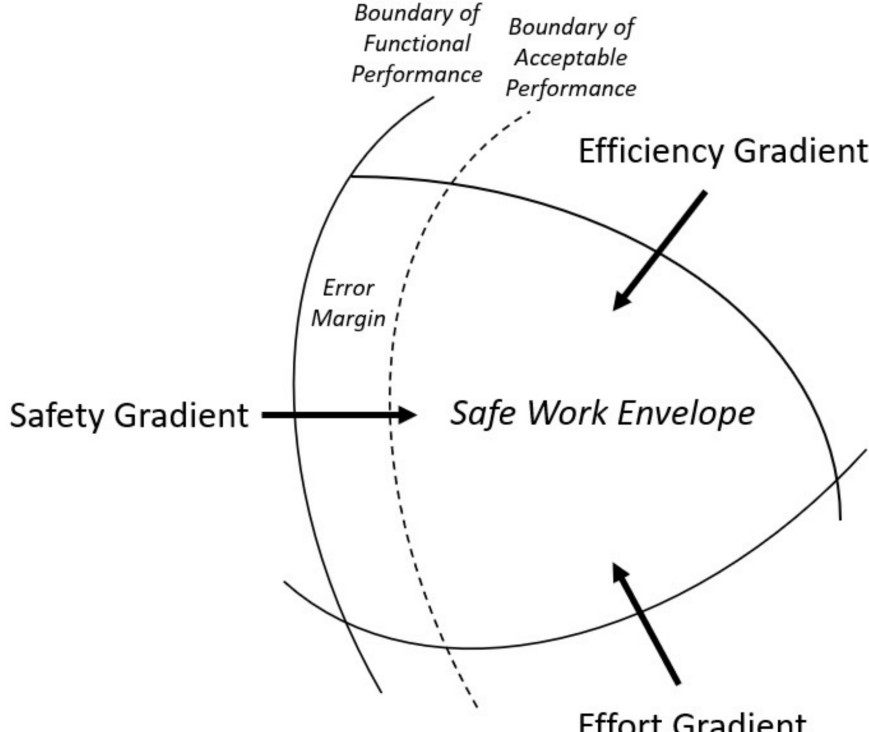

**Figure 1.** Rasmussen's dynamic risk modeling framework. Reproduced from Rasumussen [16].

Rasmussen's dynamic risk model is useful to understand how electrical safety incidents occur. For instance, a common feature of the electrical industry is the concept of work pressures that exert effects on frontline activities. A key feature of Rasmussen's model is the powerful top-down and multilayered influences that shape frontline work activities and the safety of work. Again, electrical safety is affected by forces and pressures that operate at different levels, ranging from regulators, to unions, to organizational stakeholders such as managers and peers. Furthermore, as electrical contractors often have direct contact with customers [13], there can be direct and marked effects on safety behaviors. Finally, the dynamic nature of electrical work, whereby safety boundaries (actual or perceived) can rapidly shift, and priorities swiftly change [17], means that a systems approach that takes into account dynamic influences such as these is appropriate.

Following this background and contextual setting, we describe our approach to conducting the electrical safety human factors scoping review. We show the process used to identify the environmental, individual, team, organizational, and macro factors that shape human performance in the electrical industry. Ideas from Rasmussen's model are applied to organize our findings and provide structure to our discussion. Practically, a comprehensive review of the main research databases was undertaken, including Scopus, Web of Science, and Google Scholar. Appendix A shows a summary of our initial literature search strategy.

## 2. Materials and Methods

### 2.1. Search Strategy

Two independent researchers (authors two and three on this paper) conducted the literature search using a structured string developed collaboratively by the team. One researcher targeted the Scopus database and the other targeted the Web of Knowledge database. Google Scholar was also used as a final step to find any additional literature missed in the first step. Appendix A shows the search strategy in detail.

Only the most recent (year 2000 onwards) general electrical industry human factors research was considered as being within the scope of this review. Engineering or other technical discipline papers relating to electrical safety were excluded. Exclusion criteria included studies that did not explore safety, electrical work, and human factors in combination. Conference proceedings without a clear methodology and results were excluded. Papers with their full text not available were also excluded, as were papers with languages other than English.

Using these search terms (see Appendix A), a total of 43 articles were identified. Following the removal of 25 articles deemed out of scope, irrelevant, or of questionable quality, 18 articles were included in the final review. These articles are asterisked in our references list.

### 2.2. Research Quality

An independent researcher (co-author two) reviewed each selected paper for quality using the Mixed Methods Appraisal Tool (MMAT) for quantitative studies [18], and the Critical Appraisal Skills Program tool for qualitative studies [19]. Studies that used both quantitative and qualitative methods were appraised using both tools. It is discouraged to calculate an overall score using the MMAT and the CASP [18,19]. Overall, the quality of the studies was generally good (see Appendix B). Notably, ethics statement information was not included in any of the reviewed qualitative studies, which is a significant shortfall and should be rectified in future research.

### 2.3. Analysis

Papers were initially summarized for information that could be used to inform this review. Next, the papers were reviewed in detail and specific human factors were identified. These contributors are shown in Table 1.

**Table 1.** Summary of human factors extracted from reviewed articles.

| Source | Human Factors |
| --- | --- |
| Austin et al. [13] | Importance of keeping to work schedule, protecting the public from harm, training apprentices in safe working practices, customer satisfaction, reputation protection, right person/right equipment/right job, safety attitudes, old/aging equipment, cheap or low-quality materials, preventative maintenance schedule, poor past workmanship (people without training or poor work quality), hot or dangerous machinery, other chemicals onsite/in proximity, working at heights, working in confined spaces, working in dynamic and distracting environments (e.g., construction site), poor housekeeping, weather conditions, other trades' knowledge and motivation of safety, other trades' ridicule or social pressure to work unsafely, quality of between-trade communication, customer interactions and distractions, time pressure (customers, employers, supervisors), fatigue due to long working hours, stricter safety regulations, quality work procedures and guidelines, public electrical safety campaigns, presence and availability of safety inspectors, lack of preparation when having to work live (expectation it will be dead), customer or engineers' decision to not shut down live equipment, electricians' own decision that de-energizing would be inconvenient, troubleshooting or testing live equipment, working out of hours to de-energize introduces new risks (lighting, etc.), availability of PPE, inadvertent re-energizing of deactivated equipment by others |
| Baby et al. [20] | Safety climate, personal stress, social support, job stress, self-esteem |
| Basahel [21] | Senior manager safety leadership style |
| Börner and Lassowski [22] | Fear of negative repercussions for reporting electrical incidents, absence of recognition for safe working, lack of employee involvement in decision making, relationship quality with peers, senior manager safety leadership style, internal organizational communication quality, effectiveness of job planning and resources |
| Castillo-Rosa et al. [23] | Age (<25 and over 65 more likely), experience (<1 year experience) |
| Chan et al. [24] | Workers' personal safety attitude, effectiveness of safety procedures, management safety commitment |
| Huang et al. [25] | Organizational safety climate, group-level safety climate |

**Table 1.** *Cont.*

| Source | Human Factors |
|---|---|
| Janackovic et al. [26] | Incident investigation quality, maintenance rate, application of safe work procedures, assessment/verification of work knowledge, communication between employees, training plan and implementation, inspection and auditing program, availability of resources for safety improvement programs, collection and analysis of safety information |
| Jooma et al. [27] | Risky but commonplace maintenance practices, adequacy of PPE, adherence to PPE requirements in arc flash situations, incorrect implementation of LOTO procedure, changes in work scope unrecognized, vague maintenance work procedures, aging equipment prone to failure, safety in design to retrofit or improve obsolete equipment |
| Kowalski-Trakofler and Barrett [9] | Electrical component failure, did not recognize the hazard of working live, choice to work live, experience (laborers who were inexperienced), technical officers who are experienced, fatigue, not wanting to inconvenience the customers, production pressure |
| Mobarak and Alshehri [14] | Insufficient maintenance, inadequate job planning, failure to lock out tag out, electrical knowledge and capability, hazard recognition |
| Rådman et al. [28] | Failure to use PPE, fatigue and lack of concentration, haste and deadlines, lack of hazard recognition, lack of reporting safety culture |
| Rahmani et al. [29] | Fatigue, lack of PPE, PPE not used by workers |
| White et al. [30] | Risk assessment seen as costly, time consuming, repetitive, fault finding on live equipment, saving time by working live, customer inconvenience, dangerous to isolate power, safety observers are costly, PPE can make tasks impractical, length/complexity of codes of practice and guidance |

## 3. Results

The review of the electrical safety literature identified several categories of human factors. In this section, we map out and describe each of these major and minor categories.

### 3.1. Environmental Factors

#### 3.1.1. Equipment Design and Maintenance

Maintenance standards, as well as the effects of the initial design on practices, and the quality of components used in electrical devices such as switchboards, can be problematic for electrical safety [15,28]. Maintenance practices in particular cause problems because the work may be substandard or absent, resulting in fused switches that appear to be open (but are in fact shut) or other technical issues that elevate risk. Indeed, in one study carried out in the mining industry, 74% of arc flash incidents reportedly occurred during some form of maintenance or troubleshooting activity [15]. Regarding componentry, much electrical infrastructure is aging and may suffer from degraded insulation, resulting in the exposure of conductive elements. Furthermore, in a qualitative study involving electrical workers, Austin and colleagues [13] found that there are cost pressures to replace aging equipment with cheaper componentry that may unexpectedly fail through poor design.

#### 3.1.2. Weather

This environmental characteristic carries great risk to workers. For instance, during summer, with increased humidity, the risk of skin conductance and inadvertent arcing and electric shock or electrocution is increased [29]. Weather can also affect other factors such as cognitive capacity (i.e., fatigue) [31]. The positioning of electrical equipment, and inadequate shelter from the elements such as rain can also increase risk [32]. Weather conditions may also make it less likely that electrical workers don and retain their often bulky and uncomfortable PPE [32]. Finally, heat conditions may precipitate equipment failure [32].

### 3.1.3. Physical Space

Oftentimes, electrical work may be conducted in physical spaces that are not conducive to safe work practice. For instance, electrical workers may find themselves working at heights, in confined spaces, and at odd hours due to the impact on critical infrastructure during normal business periods [9,13]. Cramped or otherwise difficult workspaces can increase risk by making it harder to access componentry while maintaining effective safety behaviors. Ergonomic factors such as poor lighting when conducting work out of hours amplify risk by exposing workers to the opportunity for inadvertent contact with live equipment. Noisy or otherwise distracting environments may also increase risk. One experimental study performed in the related sector of construction (where most electrical work occurs [4]) showed that distracted construction workers detected fewer hazards, had lower risk perception, and demonstrated fewer safety behaviors [33]. In an electrical industry-specific study, Austin and colleagues [13] found that 20% of respondents said customers create distractions that can impact the risk of inadvertently working on live circuits.

### 3.2. Individual Factors

### 3.2.1. Cognitive Capacity

Mental capacity is vital for the cognitively taxing work carried out by electricians and associated workers. Electricity is an invisible energy that requires constant vigilance to manage, given it can strike without warning and small mistakes can carry deadly consequences. In a survey-based study of electrical workers in India, job stress and personal stress were both negatively associated with safety behavior and motivation [20]. In high-risk contexts, mental stress and other factors such as fatigue deplete an employees' resources to complete work safely, with higher job demands shown to reduce safety prevention and involvement practices across multiple industries [34]. Over time, increased job demands result in burnout and work disengagement, further impacting safety practice.

### 3.2.2. Safety Knowledge and Capability

One of the most important models in safety performance involves the relationship between situational factors such as safety climate and safety leadership, personal factors such as knowledge and motivation, and safety behaviors [35]. Recent research has shown that this model also applies to the electrical industry, with Basahel [21] showing that supervisory safety leadership and pre-existing safety attitudes predict safety compliance and participation via knowledge and motivation. Case studies of arc flash incidents have highlighted the role of knowledge-based errors, such as undertaking risky work practices due to a lack of underlying capability regarding risk [27]. Others suggest that the learning loops created by an organization's safety management system enable electrical knowledge to grow, resulting in safer work practices [26]. Regarding workplace experience (i.e., years in the electrical industry), evidence is mixed. Some studies suggest that either extreme (very short or very long tenure) tends to be associated with electrical safety incidents such as arc flash, whereas others show that the influence of work experience is minimal and indirect via safety attitudes and understanding of work risk [24]. Long-tenured electricians may feel overconfident in their ability to work live or otherwise mitigate electrical hazards without following company safety protocols. Short-tenured workers could be less able to recognize situations where electrical hazards are likely or may be more risk-tolerant due to a lack of field experience [36].

### 3.2.3. Threat-Related Beliefs

In a major qualitative study of electrical workers' safety-related beliefs regarding live or energized work, Austin and colleagues [13] found that in general, 85% of interviewed electrical workers were highly cognizant of the risks involved with their job. Nevertheless, most electricians interviewed said that they had or continue to work energized, and particularly the more experienced workers with 10 or more years' experience. The study of

Austin et al. [13] highlighted the role of organizational and customer factors in overriding personal threat-related beliefs regarding risk. Overall, there is only limited evidence of electrical workers' threat beliefs [9]; however, one study in the mining sector suggested that complacency (either through inattentiveness or absence of threat-related beliefs) was implicated in 10 out of 32 arc flash incidents [9]. Furthermore, more experienced electrical workers may consciously choose to work live due to the belief that they 'know what they are doing', and 'nothing (bad) had happened before' [9]. Another study showed that when electrical workers believe that a circuit is de-energized, they are less likely to wear precautionary PPE [13].

### 3.2.4. Response Efficacy

A qualitative study conducted in Australia with electrical workers by White et al. [30] indicated several findings of note. Risk assessments were considered by workers as less effective at controlling risk because they took too much time and people forgot to complete them. The study also found that electrical workers see significant benefit in having a second person present during high-risk tasks to act as an observer and a potential rescuer. PPE efficacy was considered helpful overall yet may be less effective if bulky or poorly designed, making the job more difficult or even dangerous to complete. PPE response efficacy can also interact with weather, reducing its perceived utility during hot or humid conditions. More work needs to be carried out to understand how perceptions of response efficacy of safety controls affects frontline safety practices.

### 3.2.5. Self-Efficacy

Safety-specific self-efficacy is defined as the belief that one can persist with safety controls to reduce risk in the context of barriers or challenge such as customer pressures or safety procedures that are difficult to apply. Only one electrical safety study was found that directly referenced self-efficacy in the form of over-confidence [13] which is essentially an excess of efficacy regarding one's capacity to perform hazardous work successfully. In this way, excessive self-efficacy may predispose an electrical worker to take unnecessary risks, such as shortcuts or failing to don all required PPE.

### *3.3. Team Factors*

### 3.3.1. Social Norms

The study by White and colleagues [30] was the only study identified that explicitly investigated social norms from team members, employers, and customers in shaping electrical workers' practices. Generally, electrical workers reported a positive social pressure to conduct risk assessments (except where customers felt it was unnecessary or costly). Customers may exert a negative social pressure to work live, particularly if isolating a circuit would impact on important organizational functions. Team members may also exert negative pressure regarding the use of work/safety procedures to perform high-risk tasks and utilizing codes of practice or other lengthy guidance material on the job.

### 3.3.2. Communication

A lack of information about (1) the job, (2) the state of the equipment, and/or (3) the actions or inactions of other trades and personnel onsite increases the risk of an electrical safety incident [9]. Failures in communication can result in incorrect assumptions being made, such as believing electricity has been turned off, misplaced trust in someone or something (e.g., wiring diagrams), and an inaccurate understanding or awareness of electrical risks [13]. Such misunderstandings and inaccurate shared mental models can be created through either verbal or written communication channels. Written communications, such as signs and labels on equipment, can be worn away or mislabeled, resulting in inadvertent at-risk work behaviors. Verbal communications can be impacted by the physical environment (e.g., noise), presence of PPE (making it difficult to receive messages), work

stress such as efficiency pressures, and the quality of relationships (e.g., power differentials between apprentices and qualified tradespersons).

### 3.3.3. Safety Leadership

Supervisory safety leadership has been shown to be an influential predictor of the team's perceived safety climate (e.g., [37,38]) along with influencing electrical workers' safety knowledge and motivation directly [35]. Besides general transformational safety leadership practices non-contextualized to the electrical sector, very little is known about the specific supervisory safety leadership practices that can reduce arc flash risk.

### *3.4. Organizational Factors*
### 3.4.1. Facilitating Conditions

In several studies, electrical workers reported that the presence and quality of PPE has improved considerably over the past decade [13]. However, the study by White and colleagues [30] in general electrical safety found that simple and effective risk management procedures, such as having an extra person available to help and observe, the presence of effective energized work protocols/restrictions, workable and calibrated testing equipment, and ensuring rescue equipment is available, are all organizational characteristics that inform beliefs regarding facilitating or enabling conditions for safe work. The consultation and involvement of workers in electrical safety may also be an enabler of more effective practices, with one study showing that a lack of employee involvement in organizational decision making and management of change can increase risk [22]. Another element of facilitating conditions is job planning and work allocation. Indeed, two studies in electrical safety highlighted that ineffective planning, such as allocating inexperienced people to the task, failing to provide/review existing circuity diagrams, and developing ineffective work briefings and/or information packages increases risk [9,14].

### 3.4.2. Safety Climate

Safety climate refers to the perceived priority of safety as inferred from the safety policies, procedures, and practices of managers, supervisors, and peers [39]. Only one study to date has been conducted whereby an electrical worker-specific safety climate scale has been developed [25]. This study developed both a team-level and an organization-level safety climate tool that is appropriate for the frequent solo or small-team work that electrical personnel find themselves performing. At the organizational level, management's proactivity to rectify safety issues, quality of training, quality of equipment, effectiveness of planning and work allocation, investment in electrical maintenance and supplies, and schedule flexibility were all important safety climate factors for electrical workers. At the team level, supervisors' care and concern for individual workers, encouragement to participate in safety, and production pressure were the identified safety climate factors. Safety climate perceptions were associated with near misses, recordable incidents, and vehicle incidents.

### *3.5. Macro Factors*
### 3.5.1. Customer Expectations

Multiple studies highlighted the role of external stakeholders such as clients/customers and their expectation of fast and efficient electrical work (e.g., [9,13,32]). Electrical workers appear to be particularly susceptible to pressure from external parties to work quickly and may even work live or take shortcuts, such as failing to wear all required PPE or rushing the job if a customer asks for or expects it [13]. Clearly, customers exert a strong normative or social effect over electrical workers' thinking and practice of safe work behaviors.

### 3.5.2. Safety Legislation and Regulation

Some studies conducted within the electrical industry highlighted the ability of external legislation and regulation to impact personal decision-making regarding safety practices. Working while energized was seen as strongly discouraged by regulators in

Australia [5]. Beliefs regarding the likelihood of regulator inspections and enforcement activities may also impact workers' behaviors in the electrical industry.

## 4. Discussion

Overall, this scoping review highlighted a wide range of factors implicated in electrical safety incidents. In step with contemporary thinking about occupational safety, we advocate for a deeper and multi-level examination of the factors that shape frontline electrical worker behavior. For instance, the decision to work live and risk arc flash may be driven by a complex layering of individual self-efficacy and task knowledge, team social norms for certain safety practices, supervisory and management leadership styles, and the pressures and expectations of customers to keep electricity on for essential business functions.

### 4.1. Theoretical Implications

Our review highlights the complexity of injury prevention in the electrical industry, with a range of factors outside workers' direct control. For instance, the decisions made by customers, designers, suppliers, maintainers, and managers set up situations in which electrical incidents are more or less likely. Customers routinely exert pressure on electrical workers, which often results in at-risk behaviors such as working while energized and engaging in risky shortcuts to complete the job efficiently. In interpreting the findings from this scoping review, we refer to the ideas and concepts advanced by Rasmussen [16].

#### 4.1.1. Traditional Methods for Improving Electrical Safety

Reduce Pressure Towards Efficiency. In our review, we identified that customers, managers, and even co-workers (such as other trades on a construction site) intensify the pressure to work efficiently. According to Hollnagel [40], this tension is best summarized as the 'efficiency–thoroughness trade-off', whereby workers continually seek to balance the need to prepare, monitor, and manage, versus the need to produce, deliver, and achieve. If safety is the main concern of the organization, then thoroughness will dominate (and the pressure gradient reduces and potentially inverts to become an attractor). If production is the main concern, then the pressure gradient intensifies and pushes work closer to the unsafe boundary. Thus, one way to improve arc flash safety is to reduce the pressure of the efficiency gradient. Awareness campaigns can target customers to challenge their beliefs and expectations about working live and exerting pressure on contractors, and electrical safety legislation can be updated to reflect supply chain responsibilities (e.g., targeting maintenance workers and raising standards to prevent reworked or substandard components). Nevertheless, in today's competitive and dynamic world, organizations may find this difficult to achieve in practice.

Reduce Pressure Towards Least Effort. The effort gradient suggests that workers will seek to reduce the amount of effort that they invest in tasks to achieve goals. When effort becomes overwhelming, workers collapse from fatigue and spent resources; when effort is reduced, work again migrates closer to the unsafe boundary. A solution to the effort gradient issue is to leverage worker motivation. When workers are extrinsically motivated (e.g., monetary rewards, reduced time to complete the task due to supervisor expectations), they lack job engagement and invest minimal effort to achieve safety standards; however, when they are intrinsically motivated (e.g., adopting safety goals as their own), they are willing to invest additional effort to achieve work safely [41]. Electrical contractor organizations would benefit from leadership styles (e.g., transformational safety leadership) [42] and human resource practices (e.g., high employee involvement) [43], that build intrinsic safety motivation to prevent arc flash incidents. For instance, contractors can provide workers with discretionary budgets to purchase their own suitable PPE and engage with workers through consultation to co-design innovative control measures or retrofits for aging equipment to prevent arc flash incidents.

Increase Safety Counter-Pressure. Safety culture campaigns that focus on prevention activities (e.g., STOP programs, behavioral safety, safety awareness, and hazard recognition)

are commonplace across the electrical and related industries such as construction [44]. One of the reasons why such programs tend to be ineffective or inconclusive in their impact is because they are unable to overcome the combined forces of the efficiency and effort gradients [45]. In the worst case, increasing the safety counter pressure could result in overwhelming constraints being put on electrical workers, resulting in stopped work, as evidenced by the aviation industry, when pilots were able to stall work through intentionally following all safety protocols [11]. Overall, simply increasing the strength and salience of safety counter-pressure is likely to lead to a diminishing and even negative return past a certain point.

### 4.1.2. Contemporary Methods for Arc Flash Prevention

Highlight the Boundary of Acceptable Performance. Drawing from the high-reliability and chronic unease literatures [46,47], it is apparent that electrical safety can be improved through highlighting the boundaries beyond which safety may be compromised. For instance, cultivating a sense of healthy pessimism and the expectation that what can go wrong should be identified and adequately planned for [48]. Others describe this capability as 'safety imagination'; the ability to foresee what can go wrong and highlight where and when it is likely to occur [49]. Regardless, the ability to highlight boundaries of acceptable performance likely includes a combination of worker skills and knowledge (knowing what to look for and the potential consequences), supervisory monitoring (drawing workers' attention to tasks that step over the edge of performance), and co-worker communication (persuading and convincing team members to stop work and/or reassess the situation).

Increase the Error Margin. This strategy involves expanding the boundary of acceptable performance and reducing the space in which safe work can be performed. To do so likely requires a strong pattern of social norms across an organization, which is reinforced by both co-workers and supervisors [50]. Because this boundary is perceived, without a shared mental model regarding its location and intensity, it will likely vary across individuals. Through sense-making and sense-giving activities, such as during pre-start meetings and incident investigation outcome discussions [51], leaders may be able to convince workers to increase the error margin, drawing attention to examples of at-risk behavior that step over the newly placed boundary and emphasizing the importance of a significant error margin when dealing with high-hazard electrical equipment.

Build Ability to Recover Performance. The high-reliability literature emphasizes the importance of creating skills in emergency recovery and resilient performance under stressful conditions [52]. Furthermore, recent developments in safety leadership point to the capacity of teams to adapt in response to or in anticipation of failure, by engaging in simulations and rehearsals to build the capacity to recover performance [53]. Other activities such as ensuring rescue equipment is nearby and adequate PPE is provided and worn can also increase the ability to recover performance once the acceptable performance boundary has been crossed. Finally, adequate learning and reflection is critical after instances of unsafe performance, even (and particularly) if the outcome was successfully recovered [54]. Continuous learning about the nature of performance variability and what leads to successful recovery are the hallmarks of a resilient organization [55].

Expand the Space of Safe Operations. Innovation can develop new technologies or ways of working that either eliminate hazards or remove workers from sources of harm. De-energizing live circuits is a salient example of this strategy. By working only on dead equipment, workers can undertake activities that they would not otherwise be able to, such as manipulating wires and removing protective coverings. Nevertheless, problems can occur when the mental model of the worker differs from the current state of events (e.g., the power was reactivated without warning). Eliminating contributory factors at their source often results in the expansion of the safe operating space, which in turn reduces the risk of an arc flash hazard manifesting.

*4.2. Practical Implications*

Table 2 below shows a summary of practical implications arising from this study.

**Table 2.** Summary of practical recommendations to guide arc flash prevention.

| Control Strategy | Relevant Factor(s) | Recommendations |
| --- | --- | --- |
| Reduce pressure towards efficiency | Client interactions/pressures Safety leadership | - Train managers and supervisors to protect electrical workers from client pressures and push back on unrealistic demands; link this training to WHS duties and obligations (e.g., ensuring a safe system of work) and specific stories of electrical incidents.<br>- Develop client/customer-centric awareness campaigns that highlight their role in managing electrical safety (e.g., maintenance practices, infrastructure investment, pressures and demands placed on electrical workers). |
| Reduce pressure towards least effort | Safety knowledge and motivation Self-efficacy Safety leadership | - Develop safety training packages for industry to build capability.<br>- Promote electrical industry safety leadership that fosters intrinsic safety motivation. |
| Increase safety counter-pressure | Threat-related beliefs Safety climate | - Develop and promote electrical safety media that emphasize the susceptibility and severity of incidents (using credible and influential speakers from industry); link the campaign to specific incidents that occur.<br>- Incorporate information about electrical incidents into curricula for apprentices; consider legislating these requirements.<br>- Foster a strong and positive safety climate within organizations through aligning espousals about safety priority with enacted practices, particularly at the supervisor level. |
| Highlight boundary of acceptable performance | Threat-related beliefs Cognitive capacity Safety leadership | - Develop workers' awareness of the role of stress, emotions, and mental health on risk perception (e.g., an adapted version of mental health first aid program).<br>- Institute an in-field safety leadership observation program that seeks to review work and highlight instances of actions that approach or exceed the acceptable performance boundary.<br>- Develop an organizational policy around the use of mandatory and experienced spotters/observers for tasks that carry a high risk. |
| Increase the error margin | Social norms Communication | - Incorporate sense-making and sense-giving activities around boundary identification into daily huddles and pre-start meetings (i.e., focusing on how upcoming tasks may approach the boundary of acceptable performance and what to do to mitigate the risk). |
| Build ability to recover performance | Facilitating conditions Response efficacy | - Develop an electrical safety toolkit for organizations that outlines likely emergency scenarios and how to prevent/mitigate their effects through simulations and rehearsals.<br>- Develop and promote mandatory standards for PPE where incidents are likely to occur.- Give industry-level recognition to organizations who invest in electrical safety and innovative technical solutions. |
| Expand the space of safe operations | Equipment Physical environment Weather | - Liaise with component suppliers to ensure minimum standards of quality.<br>- Legislate reliability and safety standards for critical equipment.<br>- Routinely calibrate and test equipment.<br>- Develop organizational policies around stop-work for hazardous weather conditions; provide training and information sessions to workers. |

*4.3. Limitations*

This review was not systematic, although it used a methodical approach. This review did not consider research before 2000, or literature from technical fields such as safety engineering. Nevertheless, the list developed from this review provides a sound starting point to guide future research on the topic of electrical safety and may be used as a springboard into the investigation of specific hazard areas such as arc flash. Furthermore, from this review it is unclear which human factors are more/less influential and the mechanisms by which they may affect self-protective safety behavior. However, future research will be well placed to clarify the nature of these relationships.

*4.4. Future Research Directions*

Injury prevention in the electrical industry must be multi-pronged, and work on eliminating and engineering solutions should continue. Intervention campaigns operating

at individual, team, organizational, and regulator levels could be developed and tested from this framework. For instance, worker electrical safety training that goes beyond technical tasks and includes strategies to mitigate stress, manage distractions, and push back against client pressure, combined with supervisor and management safety leadership training, and industry-level arc flash awareness (targeting specific threat- and response efficacy-related beliefs) could be an effective way to produce change.

Furthermore, research could be conducted to explore specific areas of electrical safety such as arc flash. A program of mixed methods research would be fruitful, with qualitative research elucidating specific aspects of each human factor. Follow-up quantitative research would be useful to identify industry trends and gaps in human factors optimization. Furthermore, document analysis on electrical safety incidents would be useful to identify the most common factors implicated in these events, with potential training, education, and technical interventions leading from this work.

## 5. Conclusions

Although electrical fatalities have tended to reduce in frequency over the past decade, the rate of serious and non-life-threatening injuries (e.g., shocks, burns, and shrapnel wounds) remains unchanged and in some jurisdictions, has increased. Alarmingly, electrical contractors continue to be disciplined for transgressions of electrical safety standards that may elevate the risk of arc flash, such as reversed polarity, incorrect installations of electrical componentry (against manufacturers' guidelines) and failing to prevent access to live electrical parts [56]. This scoping review identifies an array of intervention points that could inform future industry and organizational initiatives. The key conclusion from this review is that workers' behaviors are just the starting point for electrical injury prevention; regulators and organizations must work in partnership to build and implement effective educational and capability-building campaigns, reinforced through enforcement and compliance activities.

**Author Contributions:** Conceptualization, T.W.C. and R.C.F.; methodology, T.W.C.; data curation, H.M.M. and J.H.; writing—original draft preparation, T.W.C.; writing—review and editing, R.C.F., H.M.M., and J.H.; supervision, T.W.C. and R.C.F.; project administration, T.W.C.; funding acquisition, T.W.C. and R.C.F. All authors have read and agreed to the published version of the manuscript.

**Funding:** This research was funded by The Office of Industrial Relations (Electrical Safety Office, Queensland Government).

**Institutional Review Board Statement:** Not applicable.

**Conflicts of Interest:** The authors declare no conflict of interest.

## Appendix A. Search Strings Used in This Review

(((((((ALL = (human factor*)) AND ALL = (safety)) AND ALL = (electric*)) AND ALL = (psychology)) NOT ALL = (healthcare)) NOT ALL = (medical)) NOT ALL = (vehic*)) OR ALL = (arc flash)
Results: 684
((ALL = ("culture")) AND ALL = (safety)) AND ALL = (electri*)
Results: 561
(((ALL = ("human factors")) AND ALL = (electri*)) AND ALL = (safety)) NOT ALL = (nucle*)
Results: 42
(((ALL = ("leadership")) AND ALL = (electri*)) AND ALL = (safety)) NOT ALL = (nucle*)
Results: 29

## Appendix B

*Appendix B.1. Quality Assessment of Quanitative Studies*

| Quality Assessment | Baby et al. (2021) | Basahel (2021) | Castillo Rosa et al. (2017) | Chan et al. (2018) | Huang et al. (2013) | Janackovic et al. (2020) | Jooma et al. (2017) | Radman et al. (2016) | Rahmani et al. (2013) |
|---|---|---|---|---|---|---|---|---|---|
| S1. | | | | | | | | | |
| Are there clear research questions? | Yes | Yes | Yes | Yes | Yes | Yes | Yes | Yes | Yes |
| S2. | | | | | | | | | |
| Do the collected data allow the research questions to be addressed? | Yes | Yes | Yes | Yes | Yes | Yes | Yes | Yes | Yes |
| 4.1 | | | | | | | | | |
| Is the sampling strategy relevant to address the research question? | Yes | Yes | Yes | Cannot tell | Yes | Cannot tell | Yes | Yes | Yes |
| 4.2 | | | | | | | | | |
| Is the sample representative of the target population? | Yes | Cannot tell | Yes | Yes | Cannot tell | Cannot tell | Yes | Yes | Cannot tell |
| 4.3 | | | | | | | | | |
| Are the measurements appropriate? | Yes | Yes | Yes | Yes | Yes | Yes | Yes | Cannot tell | Cannot tell |
| 4.4 | | | | | | | | | |
| Is the risk of nonresponse bias low? | Yes | Yes | Yes | No | Cannot tell | Cannot tell | Yes | No | Cannot tell |
| 4.5 | | | | | | | | | |
| Is the statistical analysis appropriate to answer the research question? | Yes | Yes | Yes | Yes | Yes | Yes | Yes | Yes | Yes |

*Appendix B.2. Quality Assessment of Qualitative Studies*

| Quality Assessment | Austin et al. (2020) | Borner and Lassowski (2019) | Huang et al. (2013) | Kowalski-Trakofler and Barrett (2007) | Techera et al. (2019) | White et al. (2016) |
|---|---|---|---|---|---|---|
| **Q1.** | | | | | | |
| Was there a clear statement of the aims in the research? | Yes | Yes | Yes | Yes | Yes | Yes |
| **Q2.** | | | | | | |
| Is a qualitative methodology appropriate? | Yes | Yes | Yes | Yes | Yes | Yes |
| **Q3.** | | | | | | |
| Was the research design appropriate to address the aims of the research? | Yes | Yes | Yes | Yes | Yes | Yes |
| **Q4.** | | | | | | |
| Was the recruitment strategy appropriate to the aims of the research? | Cannot tell | Yes | Cannot tell | Yes | Yes | Yes |
| **Q5.** | | | | | | |
| Were the data collected in a way that addressed the research issue? | Yes | Cannot tell | Yes | Yes | Yes | Yes |
| **Q6.** | | | | | | |
| Has the relationship between the researcher and the participants been adequately considered? | Yes | Cannot tell | No | Yes | Cannot tell | Yes |
| **Q7.** | | | | | | |
| Have ethical issues been taken into consideration? | Cannot tell | Cannot tell | Cannot tell | Cannot tell | Cannot tell | Cannot tell |
| **Q8.** | | | | | | |
| Was the data analysis sufficiently rigorous? | Yes | Cannot tell | Cannot tell | Cannot tell | Cannot tell | Yes |
| **Q9.** | | | | | | |
| Is there a clear statement of findings? | Yes | Yes | Yes | Yes | Yes | Yes |
| **Q10.** | | | | | | |
| Will the results help locally? How valuable is the research? | Yes | Yes | Yes | Yes | Yes | Yes |

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
