# Peer review of "Shaping Frontline Practices: A Scoping Review of Human Factors Implicated in Electrical Safety Incidents"

_safety, 2021_

Round 1

Reviewer 1 Report

The authors claim in the introduction: "This scoping review aims to describe the most recent human factors arc flash and general electrical safety literature (i.e., from 2000 onwards). The purpose of this scoping review is to expand an established and integrated model of work health and safety (WHS) self-protective behavior [3], and apply it to the arc flash context."

It is questionable what the impact of this paper might be. The authors collect, classify and summarize the manifold factors. Without doubt, such an overview is helpful and needed, but that is not new. Moreover, due to application of DeJoy's model and the special search strategy, a lot of relevant literature was removed from the list of possible articles. With selection of the most recent only (from 2000 onward), wide parts of literature are neglected. Furthermore, "Engineering or other technical discipline papers ... were excluded." Last not least, the search was shrinked by the criterion "human factors". Hence, the "key conclusion from this review ... that workers' behaviors are just the starting point from arc flash prevention..." is not that much surprising. A sum of 18 articles sounds quite a lot, but concerning the wealth of pappers on that topic it is only a very small portion.

From my point of view, it would have been interesting to describe the change of view from the older, more engineering-based literature to the newer, more scientific view.

However, the main point is the generality vs. speciality: What findings are really specific concerning arc flashs? What results are not general for work under potentially hazardous environment but are specific to work with electric devices? What of these are even more specific to arc flashs and NOT to other risks, e.g. work with high voltages?

Minor points:
- page 1: "50 000 °C" instead of "50000 C" ( Personally, I would prefer to use something like "some 10 000 K" since typical arc temperatures may reach 30-50 000 K but in close vicinity of surfaces, arc temperatures around 10-15 000 K are much more typical)
- page 1: the citation "Brandt et al., 2002" is given but neither marked by [xx] nor listed in the references.
- The tables are often very large and span over more than a single page, making visibility / readability not that easy. Maybe the authors have some ideas to improve that.

Author Response

Reviewer 1 responses are in the attached file.

Reviewer 2 Report

I hand wrote a few comments on your paper while I was reading it. They are attached to this note.

I have also included a few papers that I have written.

Author Response

Reviewer 2 responses are in the attached file.

Reviewer 3 Report

As a reviewer of the article Shaping Frontline Practices to Prevent Arc Flash: A Scoping Review of Human Factors Implicated in Electrical Arc Flash Incidents, I formulated substantive and editorial comments.

Comments on the substantive content of the reviewed article:

  1. Abstract can be written in more comprehensive and focused manner.
  2. What are the unique features of this study compared to the existing works?
  3. At the end of the Introduction, the structure of the paper and what each section contains should be presented.
  4. Contributions should be highlighted in bullet points and justified
  5. Appendixs A, B1, B2 - should be discussed in more detail

Comments on editing the reviewed article:

  1. Table 1 and Table 2 require correction. In its present form they are difficult to read.

Author Response

Reviewer 3 responses are in the attached file.

Reviewer 4 Report

The paper has an important goal, to summarize the human factors that play a role in arc flash accidents, so that researchers and practitioners can build capacity to improve safety.  The authors say they look at arc flash and general electrical safety literature – since we aren’t told which of the papers reviewed focus on arc flash specifically (I didn’t want to spend time figuring it out), it’s hard to see that this is really about arc flash, rather than electrical safety more generally. I believe most of the literature is on electrical work generally, not arc flash specifically.

Fig 1: It would be better to include the yes/no arrows found in Dejoy’s model.  And in the text, rather than suggest the actor moves from appraisal to decision making, clarify that if the threat is perceived, then they decide.  Etc.  The text currently is deterministic, suggesting the actor does identify, does initiate, etc.  If you want to modify Dejoy, perhaps rather than ‘adherence’, label the arrow out of initiation to read ‘repeated behaviours’ (rather than yes) leading to ‘ingrained habit’ which is what your text suggests the final node means.  In that case, should there be an arrow from habit back to initiate behaviour? Because in that case, the behaviours seem to be routinely taken, rather than the result of cognitive effort.  If you want to stay with Dejoy’s model as is, then it would still be better to add their yes/no arrows, and make your text less deterministic.  But I like your language of ‘ingrained habit’, which differs from ‘adherence.’ 

Re Appendix B1, question 7. In the text you note none of the qualitative studies had an ethics statement. That is quite different from “did not consider ethics.”  The ‘no’s in 7 maybe should be ‘can’t tell’ with a note clarifying there was no ethics statement.  Or better, change question 7 to “Was an ethics statement provided?” where yes/no makes sense.

Table 1 and 2 both have the same heading “Summary of human factors”.  I do not understand what analysis leads from Table 1 to Table 2.  I don’t know how to interpret Table 2.  Eg, six papers are said to mention Equipment design; one of these has an X in the ‘presence of arc hazard’ column – what does that mean?  How was that article different from the other 5?  More than that, how do human factors relate to these four stages in Dejoy’s model?  The note under table 2 does not help me understand.  I do not understand the mapping – e.g, how does cognitive capacity map to ‘presence of hazard’ rather than appraisal or decision making?

Equipment is in 3.1.1 and 3.2.3., which is confusing.  And the ‘results’ seems to just be a discussion of factors. The Dejoy model  does not seem relevant in the results.

Overall, this is said to be a scoping review, but also to expand and apply an existing model – expanding a model is not typically the purpose of a scoping review.  The model is only loosely “applied” to arc flash safety, and much of the literature is broader than arc flash.  Another existing theoretical model is suddenly introduced in the discussion.  While an important topic, and I agree a scoping review to understand what we know about the role of human behaviour in arc flash incidents would be useful, the paper lacks coherence and really does not provide that.

Author Response

Reviewer 4 responses are in the attached file.

Round 2

Reviewer 1 Report

No more comments.

Reviewer 3 Report

In terms of content and editing, the introduced corrections significantly increased the quality of the article.

Reviewer 4 Report

This paper has greatly improved since the first version. The scoping review method is well done.  A lot of work has gone into this. The theory by Rasmussen provides an organizing strategy. The conclusions are thought out.  I only have one minor edit, believing the word 'not' is missing in the 2nd sentence in the Limitations section.